# Anesthetic Management during Robotic-Assisted Minimal Invasive Thymectomy Using the Da Vinci System: A Single Center Experience

**DOI:** 10.3390/jcm11154274

**Published:** 2022-07-22

**Authors:** Ahmed Mohamed, Sharaf-Eldin Shehada, Lena Van Brakel, Arjang Ruhparwar, Marcel Hochreiter, Marc Moritz Berger, Thorsten Brenner, Ali Haddad

**Affiliations:** 1Department of Anesthesiology and Intensive Care Medicine, University Hospital Essen, University Duisburg-Essen, Hufelandstr. 55, 45147 Essen, Germany; ahmed.mohamed@uk-essen.de (A.M.); marcel.hochreiter@uk-essen.de (M.H.); marc.berger@uk-essen.de (M.M.B.); thorsten.brenner@uk-essen.de (T.B.); 2Department of Thoracic and Cardiovascular Surgery, West German Heart and Vascular Center, University Hospital Essen, University Duisburg-Essen, 45147 Essen, Germany; lena.van-brakel@gmx.de (L.V.B.); arjang.ruhparwar@uk-essen.de (A.R.)

**Keywords:** thymoma, myasthenia gravis, thymectomy, minimal invasive robotic-assisted surgery

## Abstract

Background: Robotic-assisted surgery is gaining more adaption in different surgical specialties. The number of patients undergoing robotic-assisted thymectomy is continuously increasing. Such procedures are accompanied by new challenges for anesthesiologists. We are presenting our primary anesthesiologic experience in such patients. Methods: This is a retrospective single center study, evaluating 28 patients who presented with thymoma or myasthenia gravis (MG) and undergone minimal invasive robotic-assisted thoracic thymectomy between 01/2020–01/2022. We present our fast-track anesthesia management as a component of the enhanced recovery program and its primary results. Results: Mean patient’s age was 46.8 ± 18.1 years, and the mean height was 173.1 ± 9.3 cm. Two-thirds of patients were female (n = 18, 64.3%). The preoperative mean forced expiratory volume in the first second (FEV1) was 3.8 ± 0.7 L, forced vital capacity (FVC) was 4.7 ± 1.1 L, and the FEV1/FVC ratio was 80.4 ± 5.3%. After the creation of capnomediastinum, central venous pressure and airway pressure have been significantly increased from the baseline values (16.5 ± 4.9 mmHg versus 13.4 ± 5.1 mmHg, *p* < 0.001 and 23.4 ± 4.4 cmH_2_O versus 19.3 ± 3.9 cmH_2_O, *p* < 0.001, respectively). Most patients (n = 21, 75%) developed transient arrhythmias episodes with hypotension. All patients were extubated at the end of surgery and discharged awake to the recovery room. The first 16 (57.1%) patients were admitted to the intensive care unit and the last 12 patients were only observed in intermediate care. Postoperatively, one patient developed atelectasis and was treated with non-invasive ventilation therapy. Pneumonia or reintubation was not observed. Finally, no significant difference was observed between MG and thymoma patients regarding analgesics consumption or incidence of complications. Conclusions: Robotic-assisted surgery is a rapidly growing technology with increased adoption in different specialties. Fast-track anesthesia is an important factor in an enhanced recovery program and the anesthetist should be familiar with challenges in this kind of operation to achieve optimal results. So far, our anesthetic management of patients undergoing robotic-assisted thymectomy reports safe and feasible procedures.

## 1. Introduction

Robotic-assisted surgery increases across a range of surgical specialties. Lately, robotic-assisted thoracic thymectomy (RATT) is gaining more adoption worldwide. The evolving shift away from conventional open thoracotomy to minimally invasive techniques in thoracic surgery has changed the fundamental anesthetic concerns for such procedures. The Da Vinci Robotic Surgical System (Intuitive Surgical Inc., Sunnyvale, CA, USA) is the most commonly used platform [1]. It provides three-dimensional (3D) video imaging, in addition to a set of telemanipulated flexible effector instruments [1]. The use of robotic-assisted thoracic intervention in the surgical field represents an important innovation for minimally invasive techniques, enabling overcoming limitations of the conventional standard approaches [2].

Myasthenia gravis (MG) is a disease affecting the nicotinic acetylcholine receptor of the postsynaptic membrane of the neuromuscular junction by the formation of autoantibodies against the acetylcholine receptor causing fluctuating weakness of voluntary muscles [3]. The cause of MG is still unknown, however, some theories relay its pathology to the thymus gland and its function. Thymectomy is considered the main surgical procedure to treat patients presenting with thymoma. This procedure became a widely accepted therapeutic option in patients presenting with MG [4,5]. On the other hand, minimal access thymectomy is increasing because of its comparable efficacy and less tissue trauma to conventional open surgery with similar safety [1,4,6].

Patients scheduled for minimal invasive robotic-assisted thymectomy are complex and challenging for the anesthesiologist due to many factors resulting in difficult anesthetic management. Therefore, we aimed to report our fast-track anesthesia management as a component of the enhanced recovery program in patients presenting with thymoma or myasthenia gravis undergoing robotic-assisted thymectomy via Da Vinci and to review the primary results and complications of this management.

## 2. Material and Methods

### 2.1. Study Design and Patient Population

The present study is a retrospective single center study conducted after approval from the institutional review board (IRB) and Ethic Committee of the Medical Faculty of the University Duisburg-Essen (Ref#: 22-10571-BO). The study included patients presenting with thymoma or myasthenia gravis undergoing robotic-assisted thymectomy via the Da Vinci system over a two-year period between 01/2020 through 01/2022 at the University Hospital Essen, Germany. In total, 38 patients were included and recorded in our study database. Thereafter, data were screened for eligibility, extracted, and then evaluated. Eight patients were primarily excluded as the electronic data were not complete. In two more patients, additional diaphragmatic repair was performed and have been also excluded. Finally, data sets from 28 patients were included in this study.

### 2.2. Anesthetic Management

#### 2.2.1. Preoperative Evaluation

Neurological assessment prior to surgery was performed to evaluate the presence of active or significant symptoms of MG or to optimize the medical condition. All routine investigations for patients’ evaluation were performed preoperatively: this included preoperative chest X-ray, computed tomography scan, pulmonary function tests (spirometry), complete blood examination, and an electrocardiogram. Echocardiography was not performed routinely. All preoperatively scheduled medications, especially myasthenia gravis medications were continued till the day of surgery. Meanwhile, sedative premeditations were avoided in those patients. Patients who were on steroid therapy received an intravenous hydrocortisone dose of 100 mg preoperatively.

#### 2.2.2. Monitoring

In addition to the basic monitoring including electrocardiogram (ECG), peripheral pulse oximetry (SpO2), noninvasive blood pressure measurement (NIBP), one large intravenous bore access, end tidal carbon dioxide (CO_2_) with capnometry and capnography, an arterial line was placed for beat-to-beat blood pressure monitoring and regular blood gas analysis after anesthesia induction. A central venous catheter (CVC) was placed in the right internal jugular vein as thymectomy was performed from the left side. A urinary catheter was routinely inserted and removed on the first postoperative day.

#### 2.2.3. Anesthesia Induction and Maintenance

We performed general anesthesia with induction agents which are available for every anesthetist, with less effect on cardiopulmonary system, and inexpensive. Each patient received a standard dose of 0.003 mg/KG/BW fentanyl, 2.5−3.0 mg/KG/BW propofol and 0.6 mg/KG/BW rocuronium prior to intubation. Anesthesia was maintained with volatile agents (sevoflurane with MAC 1−1.5%). We used body heat mat and kept the operation room at 23 °C to maintain patient’s body temperature. The grade of muscle relaxants blockade was controlled by continuous measuring the train of four (TOF) ratio through muscle relaxometry during the entire procedure, where continuous infusion of a muscle relaxant was avoided. If the TOF ratio reached ≥3/4, rocuronium was re-administered in a dose of 0.1–0.15 mg/KG/BW in thymoma patients and in a reduced dose of 0.05 mg/KG/BW in MG patients.

#### 2.2.4. Pain Management

Pain management was based on a combination of intercostal block (ICB), superficial local anesthetic infiltration at the site of the Da Vinci system trocar ports, and intraoperative opioid administration. After anesthesia induction every patient received intercostal block using 0.3 mL/kg/BW of bupivacaine 0.25% in three regions: in the left 3rd intercostal space mid axillary line, in the left 5th intercostal space at the anterior- or mid-axillary line, and also in the left 5th intercostal space at the parasternal line. Additionally, local anesthetic in the subcutaneous tissue as a field block using 2−3 mL bupivacaine 0.25% at each port site of the robotic arms was infiltrated. Finally, a mean opioid dose of 0.2 ± 0.19 mg fentanyl was primarily administrated for each patient, fentanyl administration was repeated in a dose of 0.001−0.002 mg/KG/BW when patients showed clinical signs of intraoperative pain.

#### 2.2.5. Patient Position

Patient was placed left side up, at 30-degree angle with a bean bag. The left arm was placed parallel to the bed with a gel pad under the arm. The optimal patient positioning is important to enabling Da Vinci access and the smooth docking of the robotic arms. Care has to be taken to position the arm, allowing free access to the intravenous cannula and the pulse oximeter to ensure correct measurement of invasive and non-invasive blood pressure, and to prevent pressure damage and nerve injury [7].

#### 2.2.6. Airway and Hemodynamic Management

All patients received a left-sided double lumen tube (DLT). Placement of the intratracheal tube was confirmed by fiberoptic bronchoscopy. After patient positioning, one lung ventilation (OLV) was initiated prior to insertion of the trocars. CO_2_ insufflation with flow rates of 5 to 8 L/min raises the intrapleural pressure and expands the surgical field to improve the view by pushing the diaphragm down toward the abdomen. This could lead to hemodynamic instability, due to compression of the mediastinal vessels resulting in arrhythmias and hypotension. Moreover, CO_2_ insufflation besides one lung ventilation may cause hypercapnia. Protective lung ventilation was applied as follows: tidal volume 3−5 mL/KG/BW, respiratory rate 15−20 per minute, PEEP of 5 mmHg, limited inspiratory pressure (maximal Pinsp. 30 cmH_2_O). A shortened I:E ratio should be avoided, which may cause air trapping in patients with chronic obstructive pulmonary disease (COPD). Even though, hypercapnia and respiratory acidosis were tolerated during OLV as long PH values were ≥7.20. Noradrenaline was applied during the procedure to manage the hemodynamic changes. The requirements for noradrenaline could be withdrawn at the end of surgery.

#### 2.2.7. Fluid Management

A restrictive fluid management strategy was applied. Usually, intraoperative blood loss is very small with no need for blood transfusion. In case of hemodynamic instability due to CO_2_ insufflation and compression of the heart and mediastinal vessels with reduction of venous return, a volume bolus in addition to administration of vasoconstrictive agents might be required. Routinely, all patients received 500−750 mL crystalloid solution during the whole procedure.

#### 2.2.8. Postoperative Management

At the end of the procedure, patient received left-sided chest tube. All patients were extubated after recruitment of both lungs and complete recovery of the neuromuscular junction (TOF (4/4) > 0.9). All patients were transferred to the recovery room for four hours, where pain control and respiratory functions were closely monitored. Thereafter, patients were deemed to be transferred to the intensive care unit (ICU) for the first 24 h. However, due to the COVID-19 pandemic situation, ICU capacity for routine surgical cases was massively reduced. So, patients were admitted to the intermediate care unit (IMC) for monitoring. All patients received a chest X-ray postoperatively and were then transferred to the normal ward and completely discharged from the hospital within 48−96 h after surgery.

### 2.3. Statistical Analysis

Statistical analysis was performed using the SPSS-software (version 27.0. IBM Crop., Armonk, NY, USA). Continuous data were expressed as means and standard deviation (SD) or medians with the 25th−75th interquartile ranges (IQR), as appropriate. Categorical data were expressed as percentages and frequencies. Differences between hemodynamic and pulmonary state after induction of anesthesia and establishing capnomediastinum, as well as between thymoma and myasthenia gravis regarding anesthesiologic data were compared using Chi-square, Fisher’s exact, or Students *t*-test. All reported *p* values are two-sided and a value of *p* < 0.05 was considered statistically significant.

## 3. Results

### 3.1. Preoperative Data

The cohort included 28 patients who presented with thymoma (15, 53.6%) or MG (13, 46.4%) and underwent robotic-assisted thymectomy between 01/2020 and 01/2022. According to the Myasthenia Gravis Foundation of America (MGFA) [8] classification, the majority of the patients (10 of 13) were classified to be IIb level with ocular, oropharyngeal, and respiratory muscles, and fewer limb muscles affecting weakness. Patients’ mean age was 46.8 ± 18.1 years and most of them were female (18, 64.3%). Mean height and weight were 173.1 ± 9.3 cm and 77.7 ± 13.9 kg, respectively. Almost one-third of the patients (8, 28.6%) presented with a history of chronic obstructive pulmonary disease (COPD), where three of them were under bronchodilators and a total of 12 (42.9%) patients had a history of nicotine abuse. Preoperative lung function tests reported a mean FEV1 of 3.8 ± 0.7 L, FVC of 4.7 ± 1.1 L, and FEV1/FVC ratio of 80.4 ± 5.3%. Patients’ demographic characteristics and risk factors are listed in Table 1.

### 3.2. Operative Data

Operative data are summarized in Table 2. All patients received double lumen intubation using a 39Ch (9, 32.1%) or 41Ch (19, 67.9%) left-sided tube. Patients underwent elective surgery via left side thymectomy, where CO2 was applied with a mean value of 6 mmHg. A chest tube was inserted at the end of the procedure and was removed 24−48 h later. Mean OLV time was 96.8 ± 28 min and mean procedural time was 157.5 ± 33 min. A blood transfusion was not required in any of the patients and patients received a mean of 600 mL crystalloid solution. All patients were extubated at the end of the procedure with full recovery of the neuromuscular junction.

### 3.3. Intraoperative Hemodynamic and Pulmonary Data

After the creation of capnomediastinum, the peripheral oxygen saturation and mean blood pressure significantly dropped in comparison to baseline values after induction of anesthesia (95.1 ± 2% versus 97 ± 1.2%, *p* < 0.001 and 68.5 ± 7 mmHg versus 81.8 ± 7 mmHg, *p* < 0.001). Most of the patients reported different arrhythmias (75%) and developed a significant decrease in heart rate (73 ± 9 bpm versus 86 ± 8 bpm, *p* < 0.001). The central venous pressure (CVP) and the airway pressure were also increased after establishing capnomediastinum (16.5 ± 4.9 mmHg versus 13.4 ± 5.1 mmHg, *p* < 0.001 and 23.4 ± 4.4 cmH_2_O versus 19.3 ± 3.9 cmH_2_O, *p* < 0.001) as reported in Table 3.

### 3.4. Perioperative Analgesia

The mean opioid (fentanyl) dose used for induction of anesthesia was 0.3 ± 0.1 mg, regional anesthesia dose using bupivacaine 0.25% was 27.8 ± 4.5 mL and intraoperative opioid (fentanyl) dose was 0.2 ± 0.19 mg. Postoperatively, all patients were admitted to the recovery room for four hours and then to the ICU (16, 57.1%) or IMC (12, 42.9%) for pain and respiratory function monitoring, from where they were discharged to the normal ward in the morning of the first postoperative day. Postoperative excessive usage of opioids was defined if the patient received ≥15 mg Piritramide, and other non-opioid analgesics if patients received ≥4 g metamizole or ≥4 g paracetamol within the first 24 h during the ICU or IMC stay. Meanwhile, patients were considered to have excessive usage of opioids or other analgesics if they required half of the above-mentioned doses during the recovery room stay. As listed in Table 4, intra- and postoperative analgesic requirements showed no difference between patients who presented with thymoma or MG.

### 3.5. Postoperative Data

One patient developed postoperative lung atelectasis and received a NIV-therapy of 5 cmH_2_O. Four patients suffered from postoperative nausea and one of them developed additional vomiting (PONV). The mean duration of ICU or IMC stay was 22.5 ± 2.0 h and the mean hospital stay was 4.3 ± 1.2 days. One patient developed a postoperative superficial wound infection and was treated in the outpatient clinic. None of the patients develop brachial plexus injury or hoarseness of voice. Neither re-intubation, major complication nor mortality was reported in any of the patients as mentioned in Table 5.

## 4. Discussion

The main findings of the current study are: (1) Our primary experience of robotic-assist thoracic thymectomy reports safe and feasible management. (2) Intraoperative hemodynamic and pulmonary parameters show significant changes with no impact on outcomes. (3) Proper anesthetic and pain management offers a smooth recovery for patients presenting with thymoma or MG without the need for prolonged ventilation or analgesic administration. (4) Postoperative ICU stays for such patients, especially those presenting with myasthenia gravis are not mandatory.

Minimally invasive surgery, patients’ clinical assessment, minimal starvation times, and optimal fluid and pain management are important components of an enhanced recovery program [9]. Robotic surgery may have longer procedural times in comparison to open surgery. However, established programs reported faster recovery and reduced analgesic requirements [10]. This robotic-assisted surgery being new and rapidly adopting surgical technology added challenges to the anesthesiologist and to the perioperative anesthesiologic management. The effects of anesthesia, patients’ position, surgical manipulation, and one lung ventilation on patient’s hemodynamics, ventilation, and perfusion provide additional considerations and require an understanding of these changes and managing experience to avoid complications and allow optimal results. So far, studies have been performed to evaluate surgical management and the outcome of this kind of surgery, meanwhile, anesthesiologic management has yet to be taken into focus. Therefore, this retrospective study was meant to discuss our anesthesiologic pitfalls in a cohort of patients presenting with thymoma or myasthenia gravis undergoing minimal invasive robotic-assisted thymectomy, where all patients required double lumen intubation with one lung ventilation to allow left-sided thymectomy.

During the procedure, CO_2_ was routinely insufflated to allow better exposure for the surgeon, this has been reported to cause gas embolism in other types of robotic surgery [11]. However, none of the patients in the current cohort reported such complications, which could be attributed to the low insufflation rate of CO_2_ which was limited to an upper flow of 8 L/min. Intraoperative resection of a mediastinal mass may lead to compression of the heart or great vessels, which could result in significant hypotension. Hence, communication between the surgeon and the anesthesiologist during the whole surgery is mandatory, especially during manipulation of the mass to anticipate hemodynamic compromise and to relieve any pressure on the heart when hypotension occurs. In the current analysis, the rise in CVP and airway pressure after the creation of capnomediastinum were statistically significant. These values were all normalized after capnomediastinum termination and the establishment of double lung ventilation with the recruitment of both lungs. In contrast to these results, Pandey et al. examined 17 myasthenia gravis patients undergoing RATT and did not observe a significant rise in CVP and airway pressure after the creation of capnomediastinum [12]. A clear explanation for this could not be found, but one can speculate that the small cohort in both studies might affect the statistical analysis. Additionally, 75% of patients in the current study developed cardiac arrhythmias associated with a decrease in the heart rate, hypotension, and a drop in oxygen saturation. This condition was easily manageable with noradrenaline infusion and volume resuscitation. Nevertheless, these transient significant changes in all those parameters had no clinical effect on the patients’ condition and outcomes.

Pain control is an important issue in those patients, previous investigators reported that high intraoperative opioid requirements increase the risk of re-admission in patients undergoing ambulatory surgery [13,14]. Therefore, adequate perioperative pain management is essential in those patients. In the current series, excessive usage of opioids or other analgesics was mainly observed within the first four hours during the recovery room stay. The use of analgesics decreased during ICU or IMC stay within the subsequent hours and almost diminished after twenty-four hours. The induction of regional anesthesia minimizes intra- and postoperative opioid consumption [15] which is of utmost importance in high-risk patients undergoing thoracic surgery [16]. This has also been observed in our analysis, the additional administration of a long-acting regional anesthetic agent using bupivacaine allowed the reduction of opioid usage during surgery and after admission to the ICU or IMC.

Muscle relaxation is another concern regarding anesthesia management in those patients. On one hand, adequate muscle relaxation is important to prevent movement or coughing and to avoid tissue or organ damage by the Da Vinci arms, on the other hand, a residual neuromuscular blockade should be avoided. Many studies reported negative effects of residual neuromuscular blockade on a postoperative patient’s condition, which may result in a prolonged intubation time, pulmonary complications, impaired wound healing, and a prolonged stay within the ICU [17,18,19]. Myasthenia gravis patients have unpredictable responses to muscle relaxants and increased susceptibility to postoperative respiratory failure, which could lead to prolonged dependence on mechanical ventilation [20]. These patients are more vulnerable to developing pneumonia because of a weak cough reflex and the immunocompromised condition. In the current study, almost half of the patients (13, 46.4%) presented with MG, and all of them were on acetylcholinesterase inhibitors, half of them (7 patients) received additional steroids, and one patient received additional Azathioprine. Kas and colleagues reported that the incidence of pulmonary complications, especially ventilator-associated pneumonia (VAP) after thymectomy in MG patients is mostly due to prolonged mechanical ventilation [21]. Li Chen et al. demonstrated in their analysis of 96 patients with MG undergoing thymectomy that, early extubation (<6 h) was associated with improved clinical outcomes, in particular with reduced risk of postoperative pulmonary infection and reduced ICU stay in comparison to those having late extubation (>6 h) [22].

Continuous monitoring with train of four (TOF) is essential to ensure a complete recovery from the muscle relaxation prior to anesthetics termination [23]. Therefore, extubation should only be carried out after full recovery of the neuromuscular junction (i.e., TOF (4/4) > 0.9), in cases of residual neuromuscular junction blockade an antagonization using sugammadex should be considered as reported earlier [24,25,26]. Of note, antagonization of the residual block in patients on acetylcholinesterase inhibitors (pyridostigmine) might be difficult or even unsuccessful [27,28]. In our series, five of the MG patients reported a late recovery of the initial muscle relaxants dose (0.6 mg/KG/BW), in those patients, more than 100 min to reach a TOF ≥ 3 was observed. In the other eight patients, a TOF ≥ 3 was reached within 40 min after the initial induction dose, so a repetitive reduced dose of rocuronium (0.05 mg/KG/BW) was used in comparison to non-MG patients who received a rocuronium dose of 0.1−0.15 mg/KG/BW. This confirms that MG patients have large variability in response to muscle relaxants, which require continuous TOF monitoring.

Unlike other investigators who reported the use of continuous administration of short-acting opioids and muscle relaxant-blocking agents (e.g., remifentanil/fentanyl- & tracium-infusion) during the whole surgery in such patients [12], we followed a different strategy in our institution. This included a combination of intermittent administration of intermediate to long-acting opioids, muscle relaxation, and regional anesthesia agents. The repetitive dose of opioids and muscle relaxants could lead to drug accumulation and unwanted side effects. This has been managed as follows: the administration of opioids was repeated when patients showed clinical signs of intraoperative pain, and the administration of muscle relaxant agents was repeated when the train of four (TOF) reached ≥3/4.

We have also observed no difference in perioperative analgesics administration between patients with or without MG. Based on this early experience, a combination of opioid and regional anesthesia make a deep grade of muscle relaxation with a TOF ratio of (0/4) during the complete procedure unnecessary. Even without a deep grade of muscle relaxation, there was no patient movement or coughing observed, this in turn allowed fast extubation of all patients at the end of the procedure. This management advocated faster re-establishing of oral medication to prevent respiratory complications and prolonged ICU stay. It should be noted that other investigators tend to change the DLT to a single lumen endotracheal tube at the end of surgery and extubate the patients later on after ICU admission to avoid respiratory complications. In the current series, we immediately extubate patients at the end of the surgery as a component of fast-track anesthesia to an enhanced recovery program and perform NIV therapy in the recovery room if required. However, four patients required postoperative noninvasive ventilation (NIV) for a couple of hours, one of them suffered from lung atelectasis and needed NIV therapy for two days, this patient was an active smoker and presented with COPD. None of the patients developed pneumonia or needed re-intubation. This also could be attributed to the non-pathological preoperative lung functions in most of the patients, with a mean FEV1/FVC ratio of 80.4 ± 5.3%. A preoperative FEV1/FVC ratio of less than 60% has been reported as a strong indicator of postoperative respiratory complications and 30-day mortality [29].

Interestingly, none of the patients developed any complications based on the severity of the MG. This could be attributed to two factors, the first is that the MG severity was mild to moderate in almost all MG patients. The second factor is related to our strict management of muscle relaxant re-administration strategy after the initial dose for intubation followed by meticulous TOF monitoring, in addition to the combination of sufficient analgesics using opioids and regional anesthesia to avoid complications.

In general, patients presenting for robotic-assisted thymectomy even in the presence of myasthenia gravis can be safely anesthetized without the need for prolonged mechanical ventilation or ICU stay. It is possible to avoid or reduce the repetition dose of neuromuscular blocking agents and it could be carefully monitored when used. The additional regional anesthesia seems to play an important role in this manner. The main anesthetic skills required for this type of surgery are similar to open thoracic surgery. However, an anesthetist should have experience in thoracic anesthesia, lung isolation techniques, one lung ventilation, and management of the related hemodynamic and pulmonary changes during the procedure, to avoid complications.

## 5. Limitation

The present study was performed at a single tertiary care medical center with a relatively small cohort. However, it represents our preliminary results of robotic-assisted thoracic thymectomy using the Da Vinci system in patients with thymoma or myasthenia gravis. Another limitation is the nature of the study being retrospective and the absence of a control group managed with a different surgical and anesthetic regime.

## 6. Conclusions

Robotic-assisted surgery is gaining popularity in many surgical fields. The current anesthesiologic management is safe and feasible, it reduces postoperative pain and allows for short hospital stays and a high level of patient satisfaction. Fast-track anesthesia is based on the core practice of thoracic anesthesia, it is an important factor in an enhanced recovery program and the anesthetist should be familiar with challenges in this kind of operation to achieve optimal results. A prospective comparison of this anesthetic regime with a control group managed with another regime is warranted.

## Figures and Tables

**Table 1 jcm-11-04274-t001:** Preoperative demographic data.

Variable	Patients (n = 28)
Demographics	
Age, years	46.8 ± 18.1
Gender, female	18 (64.3)
Height, cm	173.1 ± 9.3
Weight, kg	77.7 ± 13.9
**Risk factors & comorbidities**	
Arterial hypertension	10 (35.7)
Diabetes mellitus	2 (7.1)
Hyperlipidemia	5 (17.9)
Pulmonary hypertension	0
Preoperative creatinine level, mg/dL	0.93 ± 0.1
Impaired kidney function, GFR (<90)	9 (32.1)
Prior stroke	1 (3.6)
**Respiratory related pathologies**	
COPD	8 (28.6)
Preoperative bronchodilators	3 (10.7)
Active smoker	2 (7.1)
Previous smoker	10 (35.7)
**Preoperative Lung functions**	
FEV1, liter	3.8 ± 0.7
FVC, liter	4.7 ± 1.1
FEV1/FVC, %	80.4 ± 5.3
**Pathology**	
Thymoma without MG	15 (53.6)
Myasthenia gravis	13 (46.4)
MG with Thymic hyperplasia	4
MG without Thymoma or Thymic hyperplasia	9
Severity of myasthenia gravis	
MGFA IIa	2
MGFA IIb	10
MGFA III	1
**Therapy of myasthenia gravis (n = 13)**	
Acetylcholinesterase inhibitor	13/13
Steroid therapy	7/13
Azathioprine	1/13
**Cardiac pathologies**	
NYHA III-IV	1 (3.6)
Impaired LVEF, (EF < 50%)	0
Preoperative cardiac arrhythmia	0
Heart valvular pathology	4 (14.3)
**Anesthesia and operation risk scores**	
ASA score 1	2 (7.1)
ASA score 2	16 (57.1)
ASA score 3	10 (35.7)
ARISCAT score	36 ± 15.4
MICA score	0.3 ± 0.2

Data are presented as mean ± SD or number (%) or median (interquartile range). BMI = body mass index; GFR = Glomerular filtration rate, COPD = chronic obstructive pulmonary disease, FEV1 = forced expiratory volume; FVC = forced vital capacity, MGFA = Myasthenia Gravis Foundation of America, NYHA = New York Heart Association functional classification; LVEF = left ventricular ejection fraction; ASA = American Society of Anesthesiologist class; ARISCAT = Assess respiratory risk in surgical patients in Catalonia; MICA = Myocardial infarction and cardiac arrest.

**Table 2 jcm-11-04274-t002:** Operative data.

Variable	Patients (n = 28)
Respiratory and hemodynamic management	
Double lumen intubation, using 39Ch tube	9 (32.1)
Double lumen intubation, using 41Ch tube	19 (67.9)
One lung ventilation time, min	96.8 ± 28
Noradrenalin requirement, μg/h	23.4 ± 4.4
Fluid management, mL (Crystalloid)	600 (500−750)
Foreign blood transfusion, units	0
Immediate extubation at the end of procedure	28 (100)
Recovery time from muscle relaxants after induction of anesthesia (TOF ≥ 3/4)	
Thymoma (30 min)	15 (53.57)
Myasthenia Gravis (40 min)	8 (28.57)
Myasthenia Gravis (100 min)	5 (17.8)
**Operative details**	
Elective procedure	28 (100)
Left side thymectomy	28 (100)
CO_2_ application, mmHg	6 (5−8)
Left side chest tube insertion	28 (100)
Procedural time, min	157.5 ± 33

Data are presented as mean ± SD or number (%) or median (interquartile range). CO_2_ = Carbon dioxide. TOF = Train of four.

**Table 3 jcm-11-04274-t003:** Intraoperative hemodynamic and pulmonary changes.

Variable	Preoperative Variable (Awake, n = 28)	After Induction of Anesthesia/Intubation (n = 28)	After Creation of Capno-Mediastinum (n = 28)	*p*-Value
Heart rate, bpm	70 ± 10	86 ± 8	73 ± 9	<0.001
Heart arrhythmia	0	0	21 (75)	<0.001
MAP, mmHg	104 ± 12	81.8 ± 7.0	68.5 ± 7.0	<0.001
CVP, mmHg	--	13.4 ± 5.1	16.5 ± 4.9	<0.001
Pinsp., cmH_2_O	--	19.3 ± 3.9	23.4 ± 4.4	<0.001
SpO2, %	95.8 ± 1.7	97 ± 1.2	95.1 ± 2.0	<0.001

Data are presented as mean ± SD or number (%) or median (interquartile range). MAP = mean arterial pressure; SpO2 = peripheral oxygen saturation, HR = heart rate, Pinsp = the inspiratory pressure; CVP = central venous pressure.

**Table 4 jcm-11-04274-t004:** Perioperative analgesic data.

Variable	All Patients (n = 28)	Thymoma (n = 15)	Myasthenia Gravis (n = 13)	*p*-Value
Analgesia for induction of anesthesia				
Opioid (fentanyl), mg	0.3 ± 0.1	0.29 ± 0.1	0.28 ± 0.05	0.572
**Regional anesthesia prior to surgery**				
Bupivacaine, (mL) (2.5 mg/mL)	27.8 ± 4.5	23.3 ± 3.0	23.5 ± 5.1	0.905
**Intraoperative analgesia**				
Opioid (fentanyl), mg	0.2 ± 0.19	0.2 ± 0.15	0.2 ± 0.18	0.935
**Postoperative analgesia**				
Admission to the recovery room (4 h)	28 (100)	15 (100)	13 (100)	--
Excessive usage of opioid within recovery room	11 (39.3)	7 (46.7)	4 (30.8)	0.460
Excessive usage of other analgesics within recovery room	7 (25)	4 (26.7)	3 (23.1)	1.000
Admission to the ICU (24 h)	16 (57.1)	7 (46.7)	9 (69.2)	0.276
Admission to the IMC (24 h)	12 (42.9)	8 (53.3)	4 (30.8)	0.276
Excessive usage of opioids within ICU/IMC stay	2 (7.1)	0	2 (15.4)	0.206
Excessive usage of other analgesics within ICU/IMC stay	0	0	0	--

Data are presented as mean ± SD or number (%) or median (interquartile range). ICU = intensive care unit, IMC = intermediate care.

**Table 5 jcm-11-04274-t005:** Postoperative data.

Variable	Patients (n = 28)
Respiratory complications	
Need for NIV-CPAP (within the first 24 h)	4 (14.3)
Need for NIV-BIPAP (within the first 24 h)	0
Need for reintubation	0
Postoperative atelectasis	1 (3.6)
Postoperative pneumonia	0
**Hemodynamic complications**	
Postoperative arrhythmia	0
Postoperative CPR	0
Need for noradrenalin	3 (10.7)
**General results**	
PONV	4 (14.3)
Postoperative foreign blood transfusion, units	0
ICU/IMC stay, hours	22.5 ± 2.0
In-hospital stay, days	4.3 ± 1.2
Operative mortality	0

Data are presented as mean ± SD or number (%) or median (interquartile range). NIV = non-invasive ventilation, CPAP = continuous positive airway pressure, BIPAP = bilevel positive airway pressure, CPR = cardiopulmonary reanimation, PONV = postoperative nausea and vomiting, ICU = intensive care unit.

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
