# Peer review of "Anesthetic Management during Robotic-Assisted Minimal Invasive Thymectomy Using the Da Vinci System: A Single Center Experience"

_jcm, 2022, doi:10.3390/jcm11154274_

Round 1

Reviewer 1 Report

Dear authors

Thank you for presenting your data. The field is growing and with possible new complications and challenges.

Please include the differences and analysis of different anesthesia protocols in "conventional"surgery and robotic surgery.

Please include an control group.

I suggest a bigger group of patients for a detailed analysis.

Please include possible specific complications and difficulties in improving a new technique.

Best regards

Author Response

Reviewer 1:

Thank you for presenting your data. The field is growing and with possible new complications and challenges.

  1. Please include the differences and analysis of different anesthesia protocols in "conventional" surgery and robotic surgery

Response 1: We thank the reviewer for this comments. In our single center experience, we started the Robotic assisted thymectomy program in 2020, since then we perform all thymectomy surgeries via the Da Vinci system, that’s why we do not have another cohort undergoing conventional surgery to compare. Therefore, we aimed in this article to comment on our anesthesiologic management within the past two years during such procedure and to review our primary results so far. Hence, we are not be able to provide a comparison cohort in this this study which is managed with different anesthesia protocol. We mentioned this point within the limitation section to allow the reader to have a fair overview.

Changes 1: Page 16, Lines 355-359.

***************************************************************************2. Please include an control group. I suggest a bigger group of patients for a detailed analysis.

Response 2: We thank the reviewer for this comment. We totally agree that the cohort is relatively small, however, we included all patients operated through two years period with such maneuver to be evaluated in the current study. Based on this primary results, we would like to start a prospective analysis with another center performing the surgery with conventional method to provide a prospective comparative analysis for the readers in the future. In this descriptive single center study, we performed a standardized anesthesia protocol in patients presented with thymoma or myasthenia gravis who underwent a robotic-assisted thymectomy via the Da Vinci System. As mentioned in the first comment, all patients referred to our center would be operated via this maneuver, so, a bigger group of patients would be analyzed in the future with a comparison group as recommended. We also have mentioned this within conclusion section.

Changes 2: Page 16, Lines 365-366

***************************************************************************

  1. Please include possible specific complications and difficulties in improving a new technique.

Response 3: We thank the reviewer for this comments. As suggested we would like to comment on the following points: Unlike other investigators who reported the use of continuous administration of short acting opioids and muscle relaxant-blocking-agents (e.g. remifentanil/ fentanyl- & tracium- infusion) during the whole surgery in such patients [12], we followed a different strategy in our institution. This included, a combination of intermitted administration of intermediate to long acting opioids, muscle relaxation and regional anesthesia agents. The repetition dose of opioids and muscle relaxants could lead to drug accumulation and unwanted side effects. This has been managed as follow: the administration of opioids was repeated when patients showed clinical signs of intraoperative pain, the administration of muscle relaxant agents was repeated when the train of four (TOF) reached ≥ 3/4. Therefore, the clinical assessment of pain and a continuous TOF monitoring muscle relaxation is mandatory during the whole procedure to provide a proper anesthesia and avoid complications. A good clinical experience of the performing anesthesiologist is important to safely manage these patients in such type of operations.

Another important issue is that, other investigators tend to change the DLT to single lumen endotracheal tube at the end of surgery and extubate the patients later on after ICU admission to avoid respiratory complications [12]. In the current series, we immediately extubate patients at the end of the surgery as a component of fast-track anesthesia to enhanced recovery program and perform NIV-therapy in the recovery room if required. However, 4 patients required postoperative noninvasive ventilation (NIV) for a couple of hours, one of them suffered from lung atelectasis and needed NIV therapy for 2 days, this patient was an active smoker and presented with COPD.

Changes 3: Page 14, Lines 317-324 & Page 15, Lines 330-337

Reviewer 2 Report

Thank you for asking me to review the manuscript entitled: Anesthetic Management During Robotic-Assisted Minimal Invasive Thymectomy Using the Da Vinci System: A Single Center Experience.

Although a single Center two-years experience the authors describe the anesthetic management of patients undergoing robotic thymectomy.

The manuscript is well written and needs minimal English revision (i.e. line 139 lung not lunge, line 140-141 no need of brackets  after tidal volume and respiratory rate).

I have some suggestions:

In the demographic data it could be interesting to add the severity of MG and to evaluate if this is related to complications.

Patients with Myasthenia were all without thymoma or some patients had both? I think it could be better specified.

I suggest the authors to describe their protocol of induction of general anesthesia, which agents they use and comment on this, it could be interesting for the audience.

Author Response

Reviewer 2:

  1. The manuscript is well written and needs minimal English revision (i.e. line 139 lung not lunge, line 140-141 no need of brackets after tidal volume and respiratory rate).

Response 1: We thank the reviewer for his comments. We made the changes as required.

Changes1: Please check Line 139 now line 140, Line 141 & Line 142

***************************************************************************2. In the demographic data it could be interesting to add the severity of MG and to evaluate if this is related to complications.

Response 2: We thank the reviewer for his comments. We added the severity of MG and evaluated their relation to the complications as recommended. According to the Myasthenia Gravis Foundation of America (MGFA) [8] classification, the majority of the patients (10 of 13) were classified to be IIb level with ocular, oropharyngeal, respiratory muscles and less limb muscles affecting weakness. To be noted that, none of the patients develop any complication based on the severity of the MG. This could be attributed to two factors, the first is that the MG severity was mild to moderate in almost all MG patients. The second factor is related to our strict management of muscle relaxant re-administration strategy after the initial dose for intubation followed with meticulous TOF monitoring, in addition to the combination of sufficient analgesics using opioids and regional anesthesia to avoid complication.

Changes: Page 7, lines 177-179, Please check Table 1 & Page 15, lines 341-345

***************************************************************************3. Patients with Myasthenia were all without thymoma or some patients had both? I think it could be better specified.

Response 3: We thank the reviewer for his comments. Four out of the 13 MG had additionally Thymoma hyperplasia. This has been described in details in table 1.

Changes 3: Please check table 1

***************************************************************************4. I suggest the authors to describe their protocol of induction of general anesthesia, which agents they use and comment on this, it could be interesting for the audience.

Response 4: We thank the reviewer for his comment. We actually described our protocol of induction of general anesthesia in details within the methodology section.

Changes: Please check page 4, Lines 105-114

Round 2

Reviewer 1 Report

Thank you!

There are some limitations in the study design whit a potential for further investigations but I think that as a clinical study this is useful and interesting.